# Inhibitory Actions of Clinical Analgesics, Analgesic Adjuvants, and Plant-Derived Analgesics on Nerve Action Potential Conduction

Eiichi Kumamoto 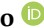

Faculty of Medicine, Saga University, 5-1-1 Nabeshima, Saga 849-8501, Japan; kumamote@cc.saga-u.ac.jp

**Definition:** The action potential (AP) conduction in nerve fibers plays a crucial role in transmitting nociceptive information from the periphery to the cerebral cortex. Nerve AP conduction inhibition possibly results in analgesia. It is well-known that many analgesics suppress nerve AP conduction and voltage-dependent sodium and potassium channels that are involved in producing APs. The compound action potential (CAP) recorded from a bundle of nerve fibers is a guide for knowing if analgesics affect nerve AP conduction. This entry mentions the inhibitory effects of clinically used analgesics, analgesic adjuvants, and plant-derived analgesics on fast-conducting CAPs and voltage-dependent sodium and potassium channels. The efficacies of their effects were compared among the compounds, and it was revealed that some of the compounds have similar efficacies in suppressing CAPs. It is suggested that analgesics-induced nerve AP conduction inhibition may contribute to at least a part of their analgesic effects.

**Keywords:** antinociception; analgesic; analgesic adjuvant; plant-derived compound; nerve conduction; sciatic nerve; compound action potential; sodium channel; potassium channel

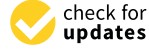



## 1. Introduction

Signal of painful stimuli applied to the skin is mainly conveyed by primary-afferent thin myelinated Aδ-fibers and unmyelinated C-fibers to the spinal cord and brain stem; the information is then transmitted to the brain by the conduction of action potentials (APs) in nerve fibers and chemical transmission at neuron-to-neuron junctions [1–4]. Acute nociceptive pain caused by tissue injury or damage is a physiological mechanism that serves to protect a person against injury, which is usually alleviated by antipyretic analgesics including non-steroidal anti-inflammatory drugs (NSAIDs) and narcotic analgesics such as opioids. On the other hand, chronic pain, which may last for a long time, such as three months or more, or occur repeatedly, is a debilitating disease accompanied by spontaneous pain, etc. and is often resistant to analgesics such as NSAIDs and opioids. Neuropathic pain, which is one type of chronic pain, results from a direct injury given to the peripheral nervous system (PNS) and damage caused in the central nervous system (CNS), and it is characterized by an excessive rise in the excitability of neurons in the vicinity of injured or damaged neuronal tissues [5]. This type of pain is alleviated by using analgesic adjuvants such as local anesthetics, antiepileptics, antidepressants, and $\alpha_2$-adrenoceptor agonists [6–15]. Although analgesics and analgesic adjuvants generally depress excitatory synaptic transmission [16–18], many of their drugs can possibly suppress nerve AP conduction, which in part contributes to their inhibitory effects on pain. Plants and their constituents are used as folk remedies to relieve pain as a drug with few side effects [19–21].

AP conduction is produced by the activation of voltage-dependent sodium and potassium channels expressed in nerve fibers. Thus, a stimulus that induces membrane depolarization, applied to a certain point on a nerve fiber, opens a sodium channel, resulting in an influx of sodium ion into the cell according to the concentration and potential gradient across the cell membrane. This leads to AP production in a self-renewing manner, which

in turn causes an outward current, i.e., membrane depolarization, to open other sodium channels at the points next to it. Such an AP production is subsided by subsequent sodium channel inactivation and potassium channel opening [22,23].

*Isolation Methods for Testing Analgesic Action on Nerve Fibers*

The study of AP in mammals is complicated by the need to dissect out individual nerves to isolate them from peripheral stimulation. For this reason, neuroscience relies on nerve extraction from relatively simple animals with conserved AP mechanisms, such as insects, reptiles, squids, or frogs (for example, refer to [23]).

AP current flowing on the surface of a nerve trunk consisting of many fibers can be measured as a compound action potential (CAP) by immersing the nerve in an isolator such as air, oil, or sucrose, and then by putting two electrodes on the nerve. CAPs, which are sensitive to tetrodotoxin (TTX), that blocks voltage-dependent sodium channels, and are fast-conducting (possibly mediated by primary-afferent thick myelinated Aα fibers), can be easily observed in the sciatic nerve trunk isolated from frogs by exposing the nerve trunk to air (known as the air-gap method). A half-peak duration of the CAP was increased by a voltage-dependent delayed rectifier potassium channel inhibitor, tetraethylammonium, without any alteration in its peak amplitude, which indicated that potassium channels are involved in CAP production [24]. Although the frog sciatic nerve exhibits both fast-conducting and slow-conducting (Aδ-fiber and C-fiber mediated) CAPs, the latter CAPs have much smaller peak amplitudes and conduction velocities than the former ones [25].

Fast-conducting CAPs recorded from the frog sciatic nerve were found to be inhibited by antinociceptive drugs in a manner dependent on their concentrations and chemical structures. Among the drugs, there are clinically used antinociceptive drugs including NSAIDs [26], many types of opioids such as tramadol [27,28], many amide- and ester-type local anesthetics [29], antiepileptics [30], antidepressants [31], dexmedetomidine (DEX; (+)-(*S*)-4-[1-(2,3-dimethylphenyl)-ethyl]-1H-imidazole, which is an $\alpha_2$-adrenoceptor agonist; [32]), and diverse kinds of antinociceptive compounds isolated from plants [33]. This entry will describe the effects of antinociceptive drugs on CAPs evoked in the sciatic nerves of frogs and argue how nerve AP conduction inhibitions produced by drugs differ among them. For comparison, the effects of antinociceptive drugs on peripheral nerve CAPs in mammals and voltage-dependent sodium and potassium channels that are involved in producing APs will also be mentioned, provided that data are available.

## 2. Actions of Analgesics on Nerve AP Conduction

### 2.1. NSAIDs

NSAIDs globally downregulate nociception through multiple mechanisms, including: inhibited production of prostaglandins from arachidonic acid by the suppression of the cyclooxygenase enzyme ([34,35]; refer to reviews [36–39]); activation of several potassium channels ([40–44]; refer to reviews [45,46]); suppression of acid-sensitive ion channels [47] and transient receptor potential (TRP) channels [48,49]; substance P depletion [50]; interaction with the adrenergic nervous system [51]; opioid receptor activation [52,53]; and cannabinoid receptor activation and increase in endocannabinoid level [54]. The idea that antinociception is produced by effects other than the suppression of cyclooxygenase is consistent with the experimental result which showed a difference between the antinociceptive and anti-inflammatory effects of NSAIDs [55].

Diclofenac, which is an acetic acid-derived NSAID, inhibited frog sciatic nerve CAPs in a partially reversible fashion; a half-maximal inhibitory concentration ($IC_{50}$) value for this activity was 0.94 mM at concentrations ranging from 0.01 to 1 mM. Like diclofenac, another acetic acid-derived NSAID, aceclofenac, which is a carboxymethyl ester of diclofenac, was shown to depress CAPs at concentrations ranging from 0.01 to 1 mM; an $IC_{50}$ value for this activity was 0.47 mM, which is less than that of diclofenac. Other acetic acid-derived NSAIDs exhibited a similar CAP inhibition, albeit to smaller extents compared with diclofenac and aceclofenac. Indomethacin at 1 mM suppressed the peak amplitude of CAP

with an extent of 38%. Acemetacin, in which the -OH group present in indomethacin is replaced with -$OCH_2COOH$, reduced CAP peak amplitudes by 38% at 0.5 mM concentration. Etodolac at 1 mM suppressed the peak amplitude of CAP with an extent of only 15%, and sulindac and felbinac at 1 mM did not affect CAP peak amplitudes [26].

Frog sciatic nerve CAP peak amplitudes were concentration-dependently reduced by fenamic acid-derived NSAIDs such as tolfenamic acid, meclofenamic acid, mefenamic acid, and flufenamic acid, the chemical structures of which resemble those of diclofenac and aceclofenac. Tolfenamic acid reduced CAP peak amplitudes with an $IC_{50}$ value of 0.29 mM at concentrations ranging from 0.01 to 0.2 mM. Meclofenamic acid, which has the -Cl group present in different numbers and positions, attached to the benzene ring of tolfenamic acid, had an $IC_{50}$ value of 0.19 mM when examined at concentrations ranging from 0.01 to 0.5 mM. Furthermore, mefenamic acid, in which the -Cl group attached to the benzene ring of tolfenamic acid is substituted with the -$CH_3$ group, reduced the peak amplitude of CAP at concentrations ranging from 0.01 to 0.2 mM (by 16% at 0.2 mM). Flufenamic acid, in which one of two -$CH_3$ groups attached to the benzene ring of mefenamic acid is absent and the other is substituted with the -$CF_3$ group, exhibited an $IC_{50}$ value of 0.22 mM, which is similar to the values exhibited by tolfenamic acid and meclofenamic acid [26]. With respect to other types of NSAIDs, frog sciatic nerve CAPs were not affected by salicylic acid-derived (aspirin; 1 mM), propionic acid-derived (ketoprofen, naproxen, ibuprofen, loxoprofen, and flurbiprofen; each 1 mM), and enolic acid-derived NSAIDs [meloxicam (0.5 mM) and piroxicam (1 mM)] [26].

The NSAIDs-induced CAP inhibition would be caused by a suppression of TTX-sensitive voltage-dependent sodium channels involved in the production of frog CAPs. Consistent with this idea, diclofenac reduced the TTX-sensitive sodium channel current amplitudes in rat dorsal root ganglion (DRG) [56] and mouse trigeminal ganglion neurons [57]. A similar sodium channel suppression caused by diclofenac has been demonstrated in myoblasts [58] and ventricular cardiomyocytes in rats [59]. Similar to diclofenac, flufenamic acid reduced the peak amplitudes of sodium channel currents in hippocampal CA1 neurons in rats [60–62]. Although the $IC_{50}$ value (0.22 mM) needed for flufenamic acid to suppress frog sciatic nerve CAPs was close to that (0.189 mM) of sodium channel suppression in hippocampal CA1 neurons in rats [62], the $IC_{50}$ value (0.94 mM) for diclofenac to inhibit CAPs was much greater than those (0.014 and 0.00851 mM in DRG neurons and myoblasts, respectively, in rats) for inhibiting sodium channels [56,58]. The rank for CAP inhibition by NSAIDs at 0.5 mM in descending order was flufenamic acid > diclofenac > indomethacin >> aspirin = naproxen = ibuprofen [26]. This order partly resembled the order for sodium channel suppression in DRG neurons (diclofenac > flufenamic acid > indomethacin > aspirin; [56]) and in cardiomyocytes (diclofenac > naproxen ≥ ibuprofen; [59]) in rats. Diclofenac at 0.3 mM reduced the TTX-resistant sodium channel current amplitudes by about 20% in trigeminal ganglion neurons in rats [63]. Flufenamic acid and tolfenamic acid at 0.1 mM reduced TTX-resistant Nav1.8 channel current amplitudes with the extents of approximately 30 and 30%, respectively [64]. Nav1.7 channel currents sensitive to TTX showed higher sensitivity to flufenamic acid and tolfenamic acid (at 0.1 mM; with the extents of approximately 60 and 70%, respectively) than TTX-resistant Nav1.8 ones [64]. Alternatively, NSAIDs inhibited the extent of increased activity produced by chemical irritation in cat corneal sensory nerve fibers; the magnitude of this inhibition was different among different types of NSAIDs [57,65]. The magnitude of NSAIDs-induced sodium channel inhibition seemed to be distinct among preparations. Concentrations needed for NSAIDs to significantly inhibit frog sciatic nerve CAPs were generally greater than those necessary for sodium channel inhibition. This result may be due to a variety of factors such as the involvement of both sodium channels and potassium channels in CAP peak amplitudes. As far as I know, it has not been reported how voltage-dependent sodium channels are affected by aceclofenac, indomethacin, etodolac, acemetacin, meclofenamic acid, and mefenamic acid. Table 1 gives a summary of the actions of NSAIDs on frog sciatic nerve fast-conducting CAPs and their $IC_{50}$ values (refer to [66] as well).

More effective NSAIDs for inhibiting frog sciatic nerve CAPs have two benzene rings connected by -NH-, and hydrophilic substituents are attached to the benzene rings (refer to Figure 1(Aa,Ba) and 3(Aa,Ba,Da) in [26] for the chemical structures of diclofenac, aceclofenac, tolfenamic acid, meclofenamic acid, and flufenamic acid). Such a chemical structure is seen in local anesthetics (refer to Section 3.1) but not in the other NSAIDs [26]. Mefenamic acid, which has a hydrophobic group attached to one of the two existing benzene rings (refer to Figure 3Ca in [26]), seemed to be efficient at suppressing CAPs, although this drug was not investigated at higher concentrations owing to its low solubility (refer to above). CAPs were efficiently suppressed by 2,6-dichlorodiphenylamine and *N*-phenylanthranilic acid, which resemble NSAIDs in that they have two benzene rings (refer to Figs. 4Aa and 4Ba in [26]); however, they are not NSAIDs. CAPs were also suppressed by the endocrine disruptor, bisphenol A, which has two benzene rings linked to a hydrophilic group such as the hydroxyl group [67].

There is much evidence in favor of the other actions of NSAIDs depending on their chemical structures. For example, the involvement of NO-cyclic GMP-potassium channels in NSAIDs-induced analgesia depends on their chemical structures [43,68]. Nonselective cation channels expressed in the pancreas in rats were suppressed by flufenamic acid and mefenamic acid, but not by indomethacin, aspirin, or ibuprofen [69]. Diclofenac and aceclofenac inhibited TRP melastatin-3 channels in a different manner [49]. Although TRP ankyrin-1 (TRPA1) channels were depressed or activated by NSAIDs, such an activity also differed in extent among NSAIDs [70]. Furthermore, a difference was observed among NSAIDs in the activities of electron transport system or mitochondrial oxidative phosphorylation that may produce adverse side effects from NSAIDs [71].

As stated above, the concentrations for NSAIDs to inhibit frog sciatic nerve CAPs are in general much greater than those necessary for the inhibition of voltage-dependent sodium channels. Such high concentrations would be possible if NSAIDs were used in close proximity to nerve fibers. At least part of the analgesic effect of NSAIDs used as a dermatological medicine may be explained by the inhibition of nerve AP conduction by suppressed voltage-dependent sodium channels [72].

Although NSAIDs are known to exhibit a tolerance effect (reduced response to a drug after its repeated use), this effect could not be explained by the NSAID-induced peripheral nerve AP conduction inhibition revealed here, because decreased frog sciatic nerve CAP through diclofenac or aceclofenac treatment recovers to almost control levels after treatment.

### 2.2. Opioids

Opioids suppress glutamatergic excitatory transmission through the activation of opioid receptors expressed in the CNS such as primary-afferent fiber central terminals, which leads to antinociception ([73–75]; refer to reviews [76,77]). Both central and peripheral terminals of primary-afferent neurons express opioid receptors; peripheral terminal opioid receptors are known to be involved in analgesia ([78–82]; refer to review [83]). Opioids also have a local anesthetic effect in the PNS. Although the perineural administration of an opioid, morphine, is reported to have no effect on CAPs in the superficial radial nerve in decerebrated cats [84], APs traveling along peripheral nerve fibers are in general inhibited by opioids. For instance, opioids including fentanyl and sufentanil reduced the peak amplitudes of CAPs evoked in peripheral nerve fibers [85] and depressed the conduction of APs in peripheral nerve fibers [86]. A CAP inhibition produced by morphine in peripheral nerve fibers in mammals was sensitive to a nonspecific opioid receptor antagonist, naloxone, indicating opioid receptor activation [87]. In support of this experimental result, binding and immunohistochemical experiments have demonstrated the localization of opioid receptors in peripheral nerve fibers in mammals [88–90].

### 2.2.1. Tramadol

Tramadol [(1*RS*; 2*RS*)-2-[(dimethylamino) methyl]-1-(3-methoxyphenyl)-cyclohexanol hydrochloride] is an orally administrated opioid in the clinical setting that acts on the CNS [91]. In animals and humans, tramadol is converted to a variety of compounds including mono-*O*-desmethyl-tramadol (M1) through *N*- and *O*-demethylation [92]; M1 is a therapeutically active drug used to alleviate pain [91]. Among cellular mechanisms for tramadol's antinociceptive effect, there is a μ-opioid receptor activation [93,94]. This idea is supported by the highest affinity of M1 among the metabolites of tramadol for cloned μ-opioid receptors. M1 inhibited excitatory transmission mediated by glutamate in spinal lamina II (substantia gelatinosa; SG) neurons, which play a pivotal part in the regulation of nociceptive transmission to the spinal dorsal horn from the periphery, leading to a reduced excitability of the neurons [95–97]. Besides such effects in the CNS, tramadol exhibits a local anesthetic effect after its intradermal application in patients ([98–100]; refer to review [101]). This result was consistent with in vivo studies which showed an inhibition of a spinal somatosensory evoked potential, which was produced by directly administrating tramadol into a rat sciatic nerve [102]. Tramadol added as an additive to local anesthetics has been reported to lengthen sensory block and analgesia duration [103].

CAPs recorded from the frog sciatic nerve were concentration-dependently reduced by tramadol with respect to their peak amplitude in the range of 0.2–5 mM in an irreversible manner [27]. A similar tramadol-induced CAP suppression has been demonstrated by other researchers in the frog [104] and rat sciatic nerves [105,106]. According to our analysis based on the Hill equation, the $IC_{50}$ value needed for tramadol-induced frog sciatic nerve CAP amplitude reduction was 2.3 mM, which was less by approximately three-fold than that (6.6 mM) reported by Mert et al. [104] for the frog sciatic nerve. Tramadol also suppressed rat sciatic nerve CAPs (37% peak amplitude reduction at 4 mM; [105]) with an extent less than that reported by Katsuki et al. [27] for frog sciatic nerve CAPs. Tramadol action in the frog sciatic nerve was unaffected by pretreating the nerve with naloxone (0.01 mM), and (D-Ala$^2$, *N*-Me-Phe$^4$, Gly$^5$-ol)enkephalin (DAMGO), which is an agonist for μ-opioid receptors, at 1 μM did not have any effect on frog sciatic nerve CAPs [27]. In contrast to tramadol, M1 was able to inhibit CAPs by much smaller extents (refer to below), although M1 has shown a higher affinity for μ-opioid receptors than tramadol whose chemical structure is similar to that of M1 [107]. These results indicate that opioid receptors are not involved in CAP inhibition produced by tramadol [27]. This idea is consistent with the observation that a spinal somatosensory evoked potential inhibition produced by tramadol in rat sciatic nerves in vivo was unaffected by naloxone [102]. Jaffe and Rowe [86] have also reported a naloxone-resistant nerve AP conduction inhibition produced by opioids.

Because the decreased frog sciatic nerve CAPs via tramadol treatment do not recover to control levels after treatment, repeated tramadol application may not have any effect, resulting in a tolerance. On the other hand, it is unlikely that the nerve AP conduction inhibition revealed here is related to tramadol's addiction, as addiction is generally thought to be due to an activation of reward centers in the brain.

Although tramadol suppresses the reuptake of noradrenaline (NA) and serotonin (5-hydroxytryptamine; 5-HT) at concentrations that are sufficient for μ-opioid receptor activation [108,109], a combination of NA and 5-HT reuptake inhibitors (desipramine and fluoxetine, respectively; each at 10 μM; refer to Section 3.3) did not affect frog sciatic nerve CAPs, indicating that CAP suppression was not mediated by NA or 5-HT reuptake inhibition [27].

It is possible that the tramadol-induced CAP inhibition is mediated by the suppression of voltage-dependent sodium and potassium channels involved in AP generation. Tramadol concentration-dependently suppressed the current amplitudes of sodium channels sensitive to TTX with an $IC_{50}$ value of 0.194 mM in DRG neuroblastoma hybridoma cell line ND7/23 cells [110]. Such a reduction also occurred in rat HEK293 cells expressing TTX-sensitive Nav1.2 channels; this activity had an $IC_{50}$ value of 0.103 mM [111]. These values were less than the $IC_{50}$ value (2.3 mM) for the inhibition of CAPs in the frog sciatic nerve [27].

Tramadol also decreased the current amplitude of delayed rectifier potassium channels (Kv3.1a type) present in NG 108-15 cells. This activity had an $IC_{50}$ value of 0.025 mM, which was much smaller than the previously mentioned 2.3 mM [112]. Such $IC_{50}$ values needed for tramadol to suppress CAPs, voltage-dependent sodium and potassium channels were greater than its reasonable serum concentration of approximately 2 μM in clinical practice [97,113].

On the other hand, CAPs in the frog sciatic nerve were unaffected by M1 (1–2 mM) in contrast to tramadol. Moreover, in the frog sciatic nerve exhibiting CAP inhibition by tramadol (1 mM; [27]), M1 (5 mM) suppressed CAP peak amplitudes with an extent of only 9%. Consistent with such smaller effects of M1, M1 (1 mM) did not block APs conducting on rat primary-afferent fibers when its effect on excitatory postsynaptic currents evoked by stimulating the dorsal root was investigated by use of whole–cell patch clamp recordings from SG neurons in spinal cord slices [96]. Interestingly, $-OCH_3$ is bound to the benzene ring for tramadol, whereas it is -OH for M1; thus, the methyl group is located on tramadol, but not on M1 (refer to Figure 5a in [27]). In conclusion, the distinction in CAP inhibition between tramadol and M1 could be explained by the difference in chemical structure.

### 2.2.2. Other Opioid-Related Compounds (Morphine, Codeine, and Ethylmorphine)

In order to know whether the structure–activity relationship between tramadol and M1 is applied to other opioids, it was examined how frog sciatic nerve CAPs are affected by morphine, codeine (where $-OCH_3$ is present instead of -OH in morphine), and ethylmorphine (where the -OH of morphine is replaced by $-OCH_2CH_3$; refer to Figure 7A in [28] for their chemical structures). Morphine at 5 mM concentration reduced the CAP peak amplitude with an extent of 15% and morphine activity was concentration-dependent; codeine (5 mM) suppressed the CAP peak amplitude with an extent of 30%. Ethylmorphine inhibited CAPs in a manner more effective than morphine and codeine (inhibition at 5 mM: 61%). Ethylmorphine activity was concentration-dependent and had an $IC_{50}$ value of 4.6 mM. The activities of morphine, codeine, and ethylmorphine were resistant to naloxone (0.01 mM). Naloxone (1 mM) itself reduced the CAP peak amplitudes by 9% while having no effect on morphine activity [28]. These results indicate that the CAP inhibitions produced by opioids were not mediated by opioid receptors, an observation similar to those of mammalian peripheral nerves [85,86,114]. On the contrary, Hunter and Frank [115] have reported a naloxone-sensitive CAP inhibition in the frog sciatic nerve. The order of the CAP peak amplitude reduction produced by opioids is ethylmorphine > codeine > morphine, indicating that the extent of CAP inhibition increases with increasing number of $-CH_2$. The observed results are consistent with the relationship between tramadol and M1 mentioned above. It is of interest to note that morphine, codeine, and ethylmorphine are absolutely different in chemical structure from tramadol and M1 [116]. Because increasing the number of $-CH_2$ groups present in an opioid increases its lipophilicity, an interaction between lipophilic opioids and ion channels has been suggested to play a crucial part in the inhibition of nerve AP conduction, as demonstrated for local anesthetics [117,118]. Structure–activity relationships similar to that of opioids have been demonstrated for other chemicals. The potency of CAP inhibition in the rat sciatic nerve decreased in the order of isopropylcocaine > cocaethylene > cocaine [119]. Interestingly, the affinity of opioids for μ-opioid receptors decreased in the order of morphine, codeine, and ethylmorphine [120]; this order is reversed to that of CAP suppression. This result supports the idea that CAP inhibition produced by opioids in the frog sciatic nerve is not thanks to opioid receptor activation.

Although opioids including morphine, codeine, and ethylmorphine are known to have a tolerance effect, this could not be explained by their inhibitory effects on nerve AP conduction, because their opioids reversibly suppress frog sciatic nerve CAPs [28].

The suppression of CAP, similar to that seen in the frog sciatic nerve, has also been shown in the mammalian peripheral nerve, albeit to varying extents. A frog sciatic nerve CAP peak amplitude reduction (by about 30%) produced by codeine (5 mM) was much

smaller than the reduction observed in the rat phrenic nerve (by approximately 70%), whereas there was not so large a distinction in morphine (5 mM) action (reduction by approximately 10%) between the two preparations. Morphine sensitivity was lower in the frog sciatic nerve compared with rabbit and guinea pig vagus nerves, whose CAP peak amplitudes were decreased with the extenst of 20–32% at 0.5 mM [87]. APs obtained by intracellular recordings from rat DRG neurons with $A\alpha/\beta$ myelinated primary-afferent fibers were also inhibited by opioids in a decreasing order of ethylmorphine, codeine, and morphine (whose $IC_{50}$ values were 0.70, 2.5, and 2.9 mM, respectively) with respect to AP peak amplitude decrease. These AP inhibitions were also resistant to naloxone (0.01 mM) [121].

AP conduction in peripheral nerve fibers is inhibited by a variety of drugs such as narcotics, antiepileptics, local anesthetics, alcohols, and barbiturates, suggesting that the drugs may interact with membrane bilayers in a nonspecific manner [122]. However, the chemical structure-dependent suppression of CAPs by opioids as described above indicates that opioids act on membrane proteins including voltage-dependent sodium and potassium channels [123]. In support of this idea, morphine inhibited the peak sodium channel currents and steady-state potassium channel currents recorded from frog sciatic nerve single myelinated fibers, resulting in the prolongation of APs [124]. The intracellular application of morphine resulted in the reduction of voltage-dependent sodium and potassium channel current amplitudes in squid giant axons [125]. Bath application of morphine reduced the peak amplitudes of sodium channel currents, sensitive to TTX, recorded from ND7/23 cells (neuroblastoma x DRG neuron hybrid cell line); this activity had an $IC_{50}$ value of 0.378 mM [110], whereas morphine at 1 mM did not affect TTX-sensitive Nav1.2 channels located in HEK293 cells [111]. Supporting the idea regarding ion channel inhibition, the opioid meperidine was prescribed for a blockade of AP conduction and in turn antinociception; meperidine depressed sodium channels in a fashion resembling that of lidocaine [126]. Table 1 gives a summary of the inhibitory actions of opioids on frog sciatic nerve fast-conducting CAPs together with their $IC_{50}$ values (refer to [66] as well).

In clinical practice, much of the pain relief from opioids is due to the administration of centrally permeable opioids into the systemic circulation, which results in opioid actions in both the PNS and CNS, leading to analgesia [127]. Opioids applied into the nerve sheath can also alleviate pain (for instance, refer to [128]). Opioids that are centrally administered can act on the PNS as well as the CNS, because opioids are moved from the brain to the periphery by an action of P-glycoproteins [129]. The subcutaneous administration of blood–brain barrier-impermeable *N*-methyl-morphine produced antinociception in an acetic acid-writhing mice model [78]. Brain-impermeable opioid loperamide subcutaneously administered showed an antinociceptive effect in a formalin test on rats [80]. Thus, nerve AP conduction inhibition produced by opioids might contribute to local antinociception after the perineural application of opioids in the periphery (for example, refer to [130]), which may lead to a direct action of opioids at high doses on peripheral nerves. Peripherally-applied codeine might have a nerve AP conduction inhibitory effect similar to morphine, because codeine is converted to morphine by *O*-demethylation in humans and animals ([131,132]; refer to review [127]).

Although opioids exhibit side effects such as tolerance and addiction, these effects appear to be mainly due to the action on synaptic transmission in the CNS rather than nerve AP conduction. This is because synaptic transmission can be plastically changed in efficacy, but this is not the case for nerve AP conduction.

## 3. Actions of Analgesic Adjuvants on Nerve AP Conduction

### 3.1. Local Anesthetics

Local anesthetics inhibit both voltage-dependent sodium and potassium channels ([117]; refer to reviews [123,133,134]). Owing to this inhibition, local anesthetics have been used to alleviate neuropathic pain in the hope of suppressing nerve AP conduction in animals [135,136] and humans [137–140], although it is possible that other effects such as

the modulation of neurotransmitter receptors, toll-like receptors, and TRP channels are also involved in analgesia [133]. Various types of local anesthetics have been reported to open TRPA1 channels located in primary-afferent neuron central terminals in the SG of the rat spinal dorsal horn [141,142] and TRPA1 and TRP vanilloid-1 (TRPV1) channels in rodent DRG neurons [143,144].

### 3.1.1. Amide-Type Local Anesthetics

Frog sciatic nerve CAPs were reversibly and concentration-dependently reduced in terms of peak amplitude by the amide-type local anesthetic lidocaine, which blocks nerve AP conduction [104–106,134]. When examined in the range of 0.1–2 mM, lidocaine's $IC_{50}$ value was 0.74 mM [28]. This value was greater than that observed for the voltage-dependent sodium channel current amplitude reduction (0.204 mM) while being smaller than the voltage-dependent potassium channel current amplitude reduction (1.118 mM) in sciatic nerve fibers in the toad *Xenopus laevis* [118]. The peak amplitudes of sodium channels resistant to TTX in rats were concentration-dependently reduced by lidocaine; an $IC_{50}$ value for this activity was 0.073 mM [145], which was 10 times less than that for the reduction of frog sciatic nerve CAP peak amplitudes. At least part of antinociception produced by systemically applied lidocaine in humans [146] may be attributed to its inhibitory effect on nerve AP conduction.

A resembling reversible CAP peak amplitude reduction was caused by another amide-type local anesthetic, ropivacaine, which exhibits a longer duration of action than lidocaine in blocking nerve conduction of APs ([147]; refer to review [148]). Ropivacaine activity was concentration-dependent over concentrations ranging from 0.01 to 1 mM; this activity had an $IC_{50}$ value of 0.34 mM [27]. The frog sciatic nerve ropivacaine-induced CAP amplitude reduction was almost comparable to that observed in rabbit vagus nerve A-fibers (approximately 30% at 0.2 mM) [149]. Frog sciatic nerve $IC_{50}$ values for lidocaine and ropivacaine, which are 0.74 and 0.34 mM, respectively, were not significantly distinct from those for fast-conducting CAP peak amplitude reduction in the rat sciatic nerve (0.28 mM for both lidocaine and ropivacaine) [150]. Moreover, an amide-type local anesthetic, prilocaine, was also shown to reversibly and concentration-dependently reduce frog sciatic nerve CAP peak amplitudes. When examined at concentrations ranging from 0.01 to 5 mM, prilocaine's $IC_{50}$ value was 1.8 mM [67].

There are levobupivacaine and its racemic, bupivacaine, which are amide-type local anesthetics; the former has a lower cardiovascular risk and CNS toxicity than the latter ([151]; refer to review [152]). CAP peak amplitudes in the frog sciatic nerve were reversibly and concentration-dependently reduced by levobupivacaine. When examined at concentrations ranging from 0.05 to 1 mM, levobupivacaine's $IC_{50}$ value was 0.23 mM [30]. This $IC_{50}$ value was close to a previously reported value (0.22 mM) for a tonic levobupivacaine-induced inhibition of frog sciatic nerve CAPs [151], and it was also close to a previously reported value (0.264 mM) for a tonic levobupivacaine-induced suppression of voltage-dependent sodium channel currents, which were recorded at the holding potential of −100 mV in GH-3 neuroendocrine cells [153]. As shown in the previous study [151], the levobupivacaine-induced CAP amplitude reduction in the frog sciatic nerve was less than that of bupivacaine (their extents at 0.5 mM were 45 and 76%, respectively [30]). This frog sciatic nerve bupivacaine activity was less than the observed values for toad *Xenopus laevis* sciatic nerve fiber sodium channels ($IC_{50}$ = 0.027 mM) [118], sodium channels sensitive to TTX in ND7/23 cells ($IC_{50}$ = 0.178 mM) [154], and rat clonal pituitary $GH_3$ cell sodium channels ($IC_{50}$ = 0.190 mM) [155]. Voltage-dependent potassium channels in sciatic nerve fibers in the toad *Xenopus laevis* were also suppressed by bupivacaine; this sensitivity ($IC_{50}$ = 0.092 mM) was less than that for sodium channels [118].

### 3.1.2. Ester-Type Local Anesthetics

As a classic ester-type local anesthetic, there is a compound (cocaine) isolated from the coca plant *Erythroxylon coca*, which has been long known to suppress nerve AP conduc-

tion ([114,156,157]; refer to review [158]). Frog sciatic nerve CAP peak amplitudes were reversibly and concentration-dependently reduced by cocaine. When examined at concentrations ranging from 0.01 to 2 mM, an $IC_{50}$ value for cocaine activity was 0.80 mM [28]. This value was similar to that obtained for lidocaine in the frog sciatic nerve (0.74 mM) [27]. The cocaine's $IC_{50}$ value was about four times larger than that observed for the rat phrenic nerve (approximately 0.2 mM) [114]. Although cocaine (40 μM) reduced mouse phrenic nerve CAP peak amplitudes by 26% [157], such a reduction in the frog sciatic nerve occurred at a concentration of approximately 300 μM [28]. There was a similar extent of CAP peak amplitude reduction by cocaine in the frog and rat sciatic nerve (30% at 0.5 mM in frogs; 40% at 0.375 mM in rats; refer to [119]). There is ample evidence for suppression of voltage-dependent sodium channels by cocaine (for example, refer to [119,159,160]); a cocaine (0.05 mM)-induced tonic (TTX-resistant) Nav1.5 channel current amplitude reduction was about 70% [160]. Cocaine and lidocaine suppressed voltage-dependent sodium channels in a competitive manner [161]. Cocaine also inhibited delayed rectifier potassium channels in central snail neurons [162].

Another ester-type local anesthetic, procaine, [163] also reduced the peak amplitudes of CAPs in the frog sciatic nerve in a reversible and concentration-dependent manner. When examined at concentrations ranging from 0.1 to 5 mM, procaine's $IC_{50}$ value was 2.2 mM [164]. This value was close to those (2–5 mM) that other researchers [165,166] reported in the same preparation and to the value obtained in the rat sciatic nerve (about 1 mM) [165]. Moreover, a ratio (2.2 mM/0.74 mM) of procaine's $IC_{50}$ value to that of lidocaine [27] in reducing frog sciatic nerve CAP amplitudes was comparable to the ratio (0.53%/0.14%) of procaine concentration necessary to block motor nerve AP conduction by 50%, to lidocaine's one in rats [167]. In addition, procaine activity in the frog sciatic nerve was 37 times less than that for voltage-dependent sodium channel current amplitude reduction in sciatic nerve fibers in the toad *Xenopus laevis* ($IC_{50}$ = 0.060 mM) [118]. Procaine also suppressed voltage-dependent potassium channels in this preparation with an $IC_{50}$ value (6.303 mM) which was larger than that obtained for sodium channels [118].

Benzocaine (ethyl 4-aminobenzoate), which is an ester-type local anesthetic, is used for not only topical anesthesia in clinical practice [168] but also amphibian anesthesia ([169]; refer to reviews [170,171]). Frog sciatic nerve CAP peak amplitudes were reversibly and concentration-dependently reduced by benzocaine. When examined at concentrations ranging from 0.01 to 2 mM, an $IC_{50}$ value for benzocaine activity was 0.80 mM (73% inhibition at 1 mM [29]). The rat sciatic nerve exhibited a similar benzocaine-induced CAP inhibition (37% inhibition at 1.3 mM [105]). The benzocaine activity was similar to those observed with cocaine and lidocaine.

Another ester-type local anesthetic, tetracaine, was also shown to reversibly and concentration-dependently reduce frog CAP peak amplitudes. When examined at concentrations ranging from 0.0005 to 0.05 mM, tetracaine's $IC_{50}$ value was 0.014 mM [32]. This value is similar to that of frog sciatic nerve fibers (0.0063 mM) as reported by Starke at al. [172] and also to that obtained for rabbit A nerve fibers (0.009 mM) [173]. On the other hand, the frog sciatic nerve tetracaine activity was 19 times smaller than that present in suppressing voltage-dependent sodium channel currents in toad *Xenopus laevis* sciatic nerve fibers ($IC_{50}$ = 0.0007 mM) [118]. In this preparation, voltage-dependent potassium channel current amplitudes were also reduced by tetracaine; an $IC_{50}$ value (0.946 mM) for this activity was much greater than for the sodium channels [118]. Tetracaine suppressed both frog sciatic nerve CAPs and toad *Xenopus laevis* sodium channels much more effectively than procaine, lidocaine, and bupivacaine.

The peak amplitudes of frog sciatic nerve CAPs were also attenuated by pramoxine that is a non-amide- and non-ester-type local anesthetic. This inhibitory action of pramoxine was concentration-dependent over concentrations ranging from 0.001 to 1 mM; this activity had an $IC_{50}$ value of 0.21 mM and subsided with a slow time course after pramoxine washout [67]. Table 1 summarizes the inhibitory actions of local anesthetics on frog sciatic nerve fast-conducting CAPs together with their $IC_{50}$ values (see [66] as well).

In clinical practice, local anesthetics locally administered to peripheral nerves may be repeatedly effective without being tolerated, because both amide-type and ester-type local anesthetics reversibly inhibit frog sciatic nerve CAPs.

### 3.2. Antiepileptics

Antiepileptics exhibit a variety of actions, such as glutamate receptor inhibition, $GABA_A$ receptor activation, and voltage-dependent sodium and calcium channel inhibition [13,174]. Antiepileptics are well-known to inhibit neuropathic pain (for example, refer to [175]). As indicated by the action on sodium channels, it is possible that the neuropathic pain alleviation is due to nerve AP conduction inhibition.

The CAP peak amplitudes of frog sciatic nerves were attenuated by lamotrigine (3,5-diamino-6-(2,3-dichlorophenyl)-1,2,4-triazine) that is a phenyltriazine derivative. Lamotriginine is reported to depress the voltage-dependent sodium channels [176] and also to attenuate a neuropathic pain accompanied by a cerebrovascular accident or diabetic polyneuropathy [7]. The sciatic nerve lamotrigine activity was in part reversible and concentration-dependent over concentrations ranging from 0.02 to 0.5 mM, and an $IC_{50}$ value for this activity was 0.44 mM [30]. This value was similar to the $IC_{50}$ value (0.641 mM at $-90$ mV) for lamotrigine to inhibit human brain type IIA sodium channels, sensitive to TTX, present in Chinese hamster ovary cells [176]. A similar CAP amplitude reduction was reported for carbamazepine (5H-dibenz[b,f]azepine-5-carboxamide; [30]), which is an iminostilbene derivative, is distinct in chemical structure from lamotrigine, and inhibits voltage-dependent sodium channels [177]. Carbamazepine has been reported to effectively attenuate trigeminal neuralgia [178,179]. Unlike the case of lamotrigine, the carbamazepine-induced CAP inhibition in the frog sciatic nerve was completely reversible. This carbamazepine activity was concentration-dependent over concentrations ranging from 0.05 to 1 mM, and an $IC_{50}$ value for this activity was 0.50 mM [30]. Carbamazepine and lamotrigine attenuated sodium channel currents in N4TG1 mouse neuroblastoma cells with similar $IC_{50}$ values [180], an observation being consistent with the experimental result that the two antiepileptics had comparable efficacies in frog sciatic nerve CAP amplitude reduction.

Oxcarbazepine (10,11-dihydro-10-oxo-5H-dibenz[b,f]azepine-5-carboxamide; [181]), which differs from carbamazepine in that it has a keto substitution at the 10,11 position of the dibenzazepine nucleus, reduced frog sciatic nerve CAP peak amplitudes; this efficacy was less than carbamazepine's one [30]. Oxcarbazepine is known to effectively relieve pains accompanied by diabetes [7] and trigeminal nerve injury [179]. Oxcarbazepine activity was partially reversible and concentration-dependent over concentrations ranging from 0.02 to 0.7 mM. The extent of CAP amplitude reduction (40%) by oxcarbazepine (0.7 mM) was slightly less than that (57%) of carbamazepine (0.7 mM). Each of lamotrigine, carbamazepine, and oxcarbazepine (all 0.5 mM) augmented the threshold required for inducing frog sciatic nerve CAPs [30]. Consistent with this experimental result, their antiepileptics reduced voltage-dependent sodium channel current amplitudes by shifting their steady-state inactivation to a more negative membrane potential [180,182,183]. Frog sciatic nerve CAP amplitude reduction produced by oxcarbazepine at 0.5 mM (20%; [30]) was much less in extent than that for the inhibition of sodium channel currents sensitive to TTX in differentiated NG108-15 neuronal cells ($IC_{50} = 3.1$ μM; [182]). Consistent with the observation that oxcarbazepine exhibited a smaller frog sciatic nerve CAP inhibition than carbamazepine, oxcarbazepine was less effective than carbamazepine in attenuating maximal electroshock-induced seizures in rats [181].

Another antiepileptic, phenytoin (hydantoin derivative, 5,5-diphenylhydantoin, which suppresses voltage-dependent sodium channels [184] and attenuates paroximal attack accompanied by trigeminal neuralgia [179]), concentration-dependently inhibited frog sciatic nerve CAPs by a small extent over concentrations ranging from 0.01 to 0.1 mM; the magnitude was only 15% at 0.1 mM [30]. The frog sciatic nerve phenytoin activity was smaller than those of rat cortical and human type IIA sodium channels, the amplitudes of

which were decreased by 60–90% by phenytoin (0.1 mM) at −60 mV [176,184]. Unlike frog sciatic nerve CAPs, voltage-dependent sodium channels were suppressed by phenytoin with an $IC_{50}$ value similar to lamotrigine in N4TG1 mouse neuroblastoma cells [182]; phenytoin, lamotrigine, and carbamazepine reportedly bound to a common site present on sodium channels in hippocampal CA1 neurons in rats [183]. The magnitude of a sensitivity of voltage-dependent sodium channels to phenytoin appeared to differ among different types of the channel. In support of this idea, the magnitudes of phenytoin activities were distinct among human Nav1.1, Nav1.2, Nav1.3, and Nav1.4 $\alpha$-subunits, sensitive to TTX, located in HEK293 cells [185]. Moreover, the properties and accessibilities of sodium channels differed between myelinated nerve fibers in frogs and rats [186].

Antiepileptics that can inhibit CAPs resemble NSAIDs in chemical structure, judging by the fact that two unsaturated six-membered rings are contained in lamotrigine, carbamazepine, and oxcarbazepine (for the chemical structures of the three antiepileptics, refer to Figures 1a, 2aA, and 2bA in [30]). Because the inhibitory effects of carbamazepine and diclofenac on voltage-dependent sodium channels occlude each other, they seem to have a common or closely related binding site [61].

In addition, frog sciatic nerve CAPs were unaffected by the other antiepileptics at concentrations as high as 10 mM [30]. Among them, there are gabapentin (1-(aminomethyl) cyclohexaneacetic acid, which has a chemical structure similar to GABA and attenuates pain persisting after herpes zoster [179]), topiramate (2,3:4,5-bis-*O*-(1-methylethylidene)-β-D-fructopyranose sulfamate, which relieves various neuropathic pains such as neuralgia involving the intercostal and trigeminal nerves [13]), and sodium valproate (2-propylpentanoic acid sodium salt, which alleviates neuropathic pain caused by diabetes [13]). The weak inhibitory effects of gabapentin and sodium valproate on frog sciatic nerve CAPs were seen with human type IIA sodium channels [176]. The human sodium channels were hardly affected by gabapentin at concentrations of less than 3 mM [176]. The antinociceptive action of gabapentin would be attributed to the fact that it binds to voltage-dependent calcium channel $\alpha_2\delta$-1 subunit, resulting in an inhibited influx of calcium ions into nerve terminals and in turn an attenuation of the release of neurotransmitters from them [187]. Gabapentin may also interrupt an interaction between *N*-methyl-D-aspartate (NMDA) receptor (which is one subtype of glutamate receptors) channels and voltage-dependent calcium channel $\alpha_2\delta$-1 subunit in postsynaptic neurons [188]. As distinct from the frog sciatic nerve, TTX-sensitive sodium channels in rat cerebellar granule cells were suppressed by topiramate; this activity had an $IC_{50}$ value of 0.0489 mM [189]. Such a difference between the sciatic nerve and cerebellar granule cells would be possibly due to a distinction in topiramate sensitivity among different sodium channel types or phosphorylated states [190]. Antinociception produced by sodium valproate and topiramate has been attributed to other mechanisms including an increase in $GABA_A$ receptor response [191,192]. An inhibition of glutamate receptors would also possibly contribute to topiramate- and lamotrigine-induced antinociceptions. This is because topiramate depresses GluK1 (GluR5) kainate receptors (one subtype of glutamate receptors) in basolateral amygdala neurons [193] and lamotrigine suppresses $\alpha$-amino-3-hydroxy-5-methyl-4-isoxazole propionate (AMPA) receptors (another subtype of glutamate receptors) in dentate gyrus granule cells in rats [194]. Table 1 gives a summary of the actions of antiepileptics on frog sciatic nerve fast-conducting CAPs and their $IC_{50}$ values (see [66] as well).

Antiepileptics that can inhibit frog sciatic nerve CAPs seemed to exhibit antinociceptive actions in a persistent pain model. Intraperitoneally administered lamotrigine, carbamazepine, and oxcarbazepine caused analgesia in the second phase of the formalin test (which reflects inflammation occurring 15–20 min after formalin injection) but phenytoin, topiramate, and sodium valproate had no effect in rats [195,196]. The antinociception produced by antiepileptics appeared to be partly due to their nerve AP conduction inhibitory actions. The plasma concentrations of lamotrigine and carbamazepine clinically prescribed to relieve epilepsy were less than 12 μM and 20–50 μM, respectively [197,198].

These values were much smaller than the $IC_{50}$ values needed for them to inhibit frog sciatic nerve CAPs.

In clinical practice, antiepileptics locally administered to peripheral nerves may be repeatedly effective without being tolerated, as the inhibitory actions of antiepileptics on frog sciatic nerve CAPs are partial but reversible.

### 3.3. Antidepressants

Antidepressants are thought to alleviate pain by activating the descending analgesic pathway composed of 5-HT- or NA-containing nerve fibers to the spinal dorsal horn from brainstem by a suppression of their neurotransmitters' reuptake [199,200], involvement of $\alpha$ adrenoceptors, $H_1$-histamine, 5-HT, opioid and muscarinic acetylcholine receptors ([11,201–204]; refer to reviews [205,206]), and the suppression of voltage-dependent calcium [207,208], NMDA receptor [209–212], and purinergic $P2X_4$ receptor (one subtype of ionotropic P2X receptors activated by ATP; [213]) channels, all of which are involved in modulating synaptic transmission. Moreover, an inhibition of neuroimmune mechanisms accompanying nerve injury may be involved in the pain alleviation produced by antidepressants [214].

Duloxetine, which is 5-HT and NA reuptake inhibitor (SNRI), ([215–217]; refer to review [218]) partially but reversibly inhibited frog sciatic nerve CAPs. Duloxetine activity was concentration-dependent over concentrations ranging from 0.001 to 2 mM, and an $IC_{50}$ value for this activity was 0.23 mM [31]. A resembling CAP inhibition was caused by fluoxetine that is a selective inhibitor of 5-HT reuptake (SSRI) ([201,202]; refer to reviews [206,219]). Fluoxetine-induced CAP peak amplitude reduction was partially but reversible, concentration-dependent over concentrations ranging from 0.05 to 5 mM, and an $IC_{50}$ value for this activity was 1.5 mM. Thus, fluoxetine was less effective in inhibiting CAPs than duloxetine [31].

Typical tricyclic antidepressants, amitriptyline and desipramine, which have tertiary and secondary amine structures, respectively [200,203,220,221], also inhibited frog sciatic nerve CAPs. Amitriptyline reduced CAP peak amplitudes over concentrations ranging from 0.001 to 1 mM, and an $IC_{50}$ value for this activity was 0.26 mM. Desipramine also showed a similar effect over concentrations ranging from 0.1 to 5 mM; this activity had an $IC_{50}$ value of 1.6 mM [31]. Thus, amitriptyline inhibited CAPs 6-fold more effectively than desipramine. Consistent with this finding, amitriptyline blocked AP conduction in the rat sciatic nerve [155]. Like tricyclic antidepressants, maprotiline, which is a tetracyclic antidepressant [220], also exhibited an inhibitory action on frog sciatic nerve CAPs in a partial but reversible fashion. Maprotiline activity was concentration-dependent over concentrations ranging from 0.2 to 5 mM, and an $IC_{50}$ value for this activity was 0.95 mM [31]. Trazodone, a $5\text{-}HT_2$ receptor antagonist and reuptake inhibitor (SARI), is a non-SNRI, non-SSRI, non-tricyclic, and non-tetracyclic antidepressant ([222–225]; refer to review [226]). Trazodone reduced frog sciatic nerve CAP peak amplitudes in a partial but reversible fashion at concentrations ranging from 0.2 to 2 mM which is the maximum soluble concentration. The magnitude of trazodone (1 mM)-induced CAP peak amplitude reduction was about 50% [31].

The antidepressants-induced CAP inhibition would be owing to an attenuation of voltage-dependent sodium channels sensitive to TTX that are involved in frog sciatic nerve CAP generation. Consistent with this idea, voltage-dependent sodium channels were inhibited by duloxetine [227,228], fluoxetine [229], amitriptyline [227,229–235], desipramine, and maprotiline [236]. Voltage-dependent sodium channels sensitive to TTX, which are located in adrenal chromaffin cells in bovines, were suppressed by amitriptyline (with an $IC_{50}$ value of 0.0202 mM), fluoxetine (0.02 mM; 62% amplitude reduction), desipramine (0.02 mM; 50% reduction), and trazodone (0.1 mM; 20% reduction; [229]). Amitriptyline also reduced the peak amplitudes of sodium channel currents in clonal pituitary $GH_3$ cells in rats; this activity had an $IC_{50}$ value of 0.0398 mM [155]. These efficacies for sodium channel inhibition were much greater than those obtained for frog sciatic nerve CAPs. Moreover, an $IC_{50}$ value

(0.0221 mM) for the inhibition of Nav1.7 channel currents, sensitive to TTX, produced by duloxetine was approximately 10 times less than that (0.23 mM) for inhibiting CAPs in the frog sciatic nerve [228]. The efficacy sequence that maprotiline more effectively depresses CAPs than fluoxetine was the same as that reported for Nav1.7 channels, where $IC_{50}$ values for inhibiting the channels by maprotiline, fluoxetine, desipramine, and amitriptyline were 0.028, 0.074, 0.024, and 0.085 mM, respectively [236]. The experimental result that amitriptyline and duloxetine inhibited frog sciatic nerve CAPs with a similar $IC_{50}$ value was the same as that reported for cardiac-type sodium channel suppression [227]. The peak amplitudes of sodium channel (possibly Nav1.8 channel)currents, resistant to TTX, in trigeminal ganglion neurons in rats were also reduced by amitriptyrine; this activity had an $IC_{50}$ value of 0.00682 mM [237]. In terms of chemical structure, typical local anesthetics have hydrophilic and hydrophobic moieties with an intermediate amide or ester linkage between them [238], while all of the antidepressants, except for trazodone, examined in the frog sciatic nerve, consist of a hydrophilic amine group and a hydrophobic moiety containing benzene rings, both of which are connected by a straight hydrocarbon chain (refer to Figure 1 in [31] for their antidepressants' chemical structures). Such chemical structures may play a pivotal role in inhibiting sodium channels. Table 1 gives a summary of $IC_{50}$ values for antidepressants to inhibit fast-conducting CAPs in the frog sciatic nerve (see [66] as well).

The antidepressants examined in the frog sciatic nerve are clinically used to alleviate chronic pain [10,11,217,218,221,239,240] and attenuate neuropathic pain in animal models. For instance, duloxetine attenuated tactile allodynia (where pain is caused by a stimulus that is normally painless) and heat hyperalgesia (where the sensitivity to painful stimuli is abnormally increased) in neuropathic pain models in rats [215]. Fluoxetine caused analgesia in diabetic neuropathic pain models produced by streptozotocin in mice [201]. Amitriptyline and desipramine effectively attenuated pain in patients suffering from diabetic neuropathy [200]. Maprotiline suppressed neuropathic pain in rats undergoing chronic constriction injury to the sciatic nerve [241]. Trazodone depressed hyperalgesia produced in chronic constriction injury models in rats [222]. The plasma concentrations of duloxetine, fluoxetine, amitriptyline, desipramine, maprotiline, and trazodone prescribed for the treatment of depression and neuropathic pain in clinical practice are 0.09–0.3, 0.3–1.6, 0.36–0.90, 0.47–1.1, 0.72–1.4, and 2.2–4.3 µM, respectively [205,228]. These concentration values were much smaller than the $IC_{50}$ values needed for them to inhibit frog sciatic nerve CAPs. The antidepressants may only produce an analgesia when locally applied to the nerve.

In clinical practice, antidepressants locally administered to peripheral nerves may be repeatedly effective without being tolerated, as the inhibitory actions of antidepressants on frog sciatic nerve CAPs are partial but reversible.

### 3.4. Adrenoceptor Agonists

The epidural and intrathecal administration of $\alpha_2$ adrenoceptor agonists such as clonidine and DEX [242] results in antinociception in animals [243–245] and humans [246]. This is possibly due to the inhibition produced by the agonists of excitatory transmission mediated by glutamate in spinal superficial dorsal horn neurons [247]. Administration of $\alpha_2$ adrenoceptor agonists and local anesthetics for the purpose of spinal anesthesia prolongs peripheral nerve conduction block in animals [248–251] and humans ([252–259]; refer to review [260]). This is possibly due to a local vessel contraction produced by the agonists, leading to decreased removal of the anesthetics from the subarachnoid space [261,262]. Furthermore, $\alpha_2$ adrenoceptor agonists attenuate nerve AP conduction and thus have a local anesthetic effect, leading to an enhanced effect of local anesthetic [263]. For instance, clonidine produced both a suppression of excitatory transmission in rat spinal SG neurons [264,265] and a blockade of peripheral nerve AP conduction [172,263,266]; the latter effect required much higher concentrations of clonidine than the former effect. Similar to clonidine, DEX reportedly inhibited excitatory transmission in rat SG neurons [267]. When

DEX or clonidine, combined with lidocaine, was intracutaneously applied into the back of guinea pigs, the local anesthetic effect of lidocaine was enhanced [268]. Local wound infiltration with DEX together with another local anesthetic, bupivacaine, more effectively attenuated pain that occurred after surgery than bupivacaine alone in humans [269]. In addition to clonidine, DEX has a possible inhibitory action on nerve AP conduction, as DEX is reported to suppress voltage-dependent sodium channels [145] (refer to below).

The CAP peak amplitudes of the frog sciatic nerve were reduced by DEX in a concentration-dependent and reversible fashion. When examined at concentrations ranging from 0.01 to 1 mM, DEX's $IC_{50}$ value was 0.40 mM [32]. The DEX activity was resistant to the $\alpha_2$-adrenoceptor antagonists, yohimbine and atipamezole ([247,270–272]; refer to reviews [273,274]), although DEX exhibited a high affinity for $\alpha_2$ adrenoceptors [242]. This result indicates no involvement of $\alpha_2$ adrenoceptors in the DEX activity [32]. CAP peak amplitude reductions were also produced by another $\alpha_2$-adrenoceptor agonist, oxymetazoline, which is more selective to $\alpha_{2A}$ than $\alpha_{2B}$ and $\alpha_{2C}$ adrenoceptors [273,275]; clonidine also presented CAP peak amplitude reductions in a manner insensitive to yohimbine. Oxymetazoline concentration-dependently and reversibly reduced the CAP peak amplitude, and an $IC_{50}$ value for this activity was 1.5 mM. Clonidine at 2 mM reduced the CAP peak amplitude by about 20% [32]. The extent of this clonidine activity was distinct from the results (CAP amplitude reduction of 80% at 0.3 mM) obtained by Starke et al. [172] using the same frog sciatic nerve, although the reason why this difference occurs is not known. In addition, a variety of adrenoceptor agonists, adrenaline, NA, phenylephrine ($\alpha_1$-adrenoceptor agonist), and isoproterenol (a $\beta$-adrenoceptor agonist) at 1 mM did not affect CAPs in the frog sciatic nerve [32]. A similar CAP peak amplitude reduction caused by clonidine has been demonstrated in the sciatic nerve in rats. Thus, CAPs derived from primary-afferent A$\alpha$ and C fibers contained in the sciatic nerve in rats were found to be suppressed by clonidine; $IC_{50}$ values in their fibers were 2.0 and 0.45 mM, respectively [266].

The $\alpha_2$-adrenoceptor agonists-induced CAP inhibition would be mediated by a suppression of voltage-dependent sodium and potassium channels that are involved in AP generation. DEX reportedly reduced the voltage-dependent sodium channel current amplitude in DRG neurons in a manner resistant to yohimbine in rats, although this type of sodium channels was insensitive to TTX [145]. The $IC_{50}$ value (0.058 mM) for this DRG neuron DEX activity in rats was approximately ten times less than that (0.40 mM) required for sciatic nerve CAP suppression in frogs. The peak amplitude of sodium channel current insensitive to TTX in rats was also reduced by clonidine; this activity had an $IC_{50}$ value of 0.26 mM [145]. The peak amplitudes of TTX-sensitive sodium channel currents in ND7/23 cells were reduced by clonidine; an $IC_{50}$ value for this reduction was 0.824 mM [154]. It has been demonstrated in NG108-15 neuronal cells that the peak amplitudes of delayed rectifier potassium channel currents were reduced by DEX; an $IC_{50}$ value for this activity was 0.0046 mM, while the peak amplitudes of sodium channel currents sensitive to TTX were attenuated by approximately 20% by DEX (0.01 mM) in a manner resistant to yohimbine [276]. These differences in drug potency may be caused by a distinction in either sodium channel types or animal species. Table 1 demonstrates the actions of adrenoceptor agonists on frog sciatic nerve fast-conducting CAPs together with their $IC_{50}$ values (refer to [66] as well).

In clinical practice, DEX administration results in producing analgesia/sedation and decreasing heart rate, cardiac output, and memory; different plasma concentrations of DEX are effective in each of these effects [277]. In patients, sedation is rapidly induced by administering at a rate of 0.2–0.7 mg·kg$^{-1}$·hr$^{-1}$ (intravenous) [242]. In intramuscular applications in cats, 40 mg·kg$^{-1}$ is the dose used usually to produce analgesia/sedation [278]. The concentrations of DEX necessary for frog sciatic nerve AP conduction inhibition are more than 1000 times higher than those of DEX used as $\alpha_2$ adrenoceptor agonist, as plasma levels of DEX in clinical use are less than 0.05 μM (refer to [277]). Therefore, the potency of DEX in blocking nerve AP conduction does not depend on the usage of DEX for analgesia/sedation. $\alpha_2$-Adrenoceptor agonists such as DEX, together with a local anesthetic,

have been applied for prolonging the duration of AP conduction block in peripheral nerve fibers [249,252–255,279]. This effect is possibly due to a local vasoconstriction leading to slowing local anesthetic absorption and/or a direct nerve AP conduction suppression produced by $\alpha_2$ adrenoceptor agonists [256]. The latter mechanism would be the $\alpha_2$ adrenoceptor agonists-induced nerve CAP inhibition, as stated above. This action becomes meaningful when their topical administration on nerves is considered but is not related to analgesia/anesthesia caused by their systemic application. Certain chemical structures of their $\alpha_2$ adrenoceptor agonists (refer to [32]) may play a crucial role in the production of nerve AP conduction blockage. DEX locally administered to peripheral nerves may be repeatedly effective without being tolerated, as DEX reversibly inhibits frog sciatic nerve CAPs.

## 4. Comparison in Nerve AP Conduction Inhibition among Analgesics and Analgesic Adjuvants

As noted from Table 1, some analgesic adjuvants had similar $IC_{50}$ values for frog sciatic nerve CAP inhibitions. For example, antidepressants had $IC_{50}$ values resembling those of some of local anesthetics, antiepileptics, and $\alpha_2$-adrenoceptor agonists. The $IC_{50}$ values of duloxetine and amitriptyline, which are 0.23 and 0.26 mM, respectively, resemble the values that ropivacaine, levobupivacaine, pramoxine, lamotrigine, carbamazepine, and DEX have (0.34, 0.23, 0.21, 0.44, 0.50, and 0.40 mM, respectively). On the other hand, the $IC_{50}$ values of fluoxetine, desipramine, maprotiline, and trazodone, which are 1.5, 1.6, 0.95, and ca. 1 mM, respectively, were comparable to those of lidocaine, cocaine, procaine, prilocaine, and oxymetazoline (0.74, 0.80, 2.2, and 1.8 and 1.5 mM, respectively). There was not a common chemical structure among the former ($IC_{50}$ values ranging from 0.2 to 0.5 mM) or latter ($IC_{50}$ values ranging from 0.7 to 2 mM) drugs, although the number of $CH_2$ in opioids having similar structures was related to the extent of CAP inhibition (refer to Section 2.2). The antidepressants had $IC_{50}$ values that were much greater than that (0.014 mM) of tetracaine. Therefore, several of the analgesic adjuvants are able to inhibit nerve AP conduction with a similar efficacy to each other.

By comparing the $IC_{50}$ values of analgesic adjuvants with those of antipyretic analgesics NSAIDs, diclofenac's $IC_{50}$ value (0.94 mM) was close to the values that lidocaine, cocaine, maprotiline, and trazodone have (0.74, 0.80, 0.95, and ca. 1 mM, respectively); the $IC_{50}$ values of aceclofenac, tolfenamic acid, meclofenamic acid, and flufenamic acid, which are 0.47, 0.29, 0.19, and 0.22 mM, respectively, were similar to the values that ropivacaine, levobupivacaine, pramoxine, duloxetine, amitriptyline, lamotrigine, carbamazepine, and DEX have (0.34, 0.23, 0.21, 0.23, 0.26, 0.44, 0.50, and 0.40 mM, respectively). The $IC_{50}$ values of NSAIDs were less than those of procaine, prilocaine, fluoxetine, desipramine, and oxymetazoline (2.2, 1.8, 1.5, 1.6, and 1.5 mM, respectively), whereas being greater than that (0.014 mM) of tetracaine. Therefore, NSAIDs could suppress nerve AP conduction with efficiencies that are similar to those of several analgesic adjuvants. In these cases, common chemical structures were not noticed among compounds with similar $IC_{50}$ values.

Aside from NSAIDs and analgesic adjuvants, narcotic analgesics, opioids, can also inhibit nerve AP conduction. Frog sciatic nerve CAP peak amplitudes were also reduced by opioids; the $IC_{50}$ values of tramadol and ethylmorphine were 2.3 and 4.6 mM, respectively; morphine and codeine (each 5 mM) attenuated CAP amplitude by 15 and 30%, respectively. The magnitudes of these opioid actions were less than those of NSAIDs and analgesic adjuvants. For instance, the $IC_{50}$ value (2.3 mM) of tramadol was greater than the values of lidocaine and ropivacaine, which are 0.74 and 0.34 mM, respectively, by 3.1 and 6.8 times, respectively [27]. Lidocaine has been previously reported in the frog sciatic nerve by other investigators to reduce CAP amplitudes with an $IC_{50}$ value of 6.6 mM, which is greater than the tramadol value by three-fold [106]. The ratio of the $IC_{50}$ value of tramadol to that of lidocaine was almost similar to that reported in the frog sciatic nerve by Katsuki et al. [27], although the $IC_{50}$ values of lidocaine greatly differed between the two studies [27,106]. The extent of a contribution of nerve AP conduction inhibition to opioids-induced analgesia

seems to be much less than other well-known cellular mechanisms including membrane hyperpolarization and decreased amount of L-glutamate released from nerve terminals in the spinal dorsal horn (for instance, refer to [73,74]). In conclusion, nerve AP conduction suppression may be a common cellular mechanism for NSAIDs and analgesic adjuvants, but not opioids, to cause analgesia.

The $IC_{50}$ values for frog sciatic nerve CAP inhibition produced by the analgesic adjuvants were similar to those in rat sciatic nerve CAPs and were in general larger than the $IC_{50}$ values for TTX-sensitive sodium channel inhibitions. This difference could be explained by several possibilities. First, CAPs are produced by not only voltage-dependent sodium channels but also potassium channels. Second, the expression of sodium channel types (Nav1.1-1.4, Nav1.6, and Nav1.7) sensitive to TTX may be different, depending on the preparations examined. Third, CAPs originate from a bundle of nerve fibers but sodium currents originate from individual cells. When the analgesic adjuvants prescribed in clinical practice act on nerve bundles, their sciatic nerve $IC_{50}$ values may be a good guide to know whether the drugs suppress nerve AP conduction in vivo, as nerve conduction is due to the activation of both voltage-dependent sodium and potassium channels. Taking into consideration the experimental result that the nociceptor-specific deletion of the gene of Nav1.7 channels sensitive to TTX leads to inhibited acute and chronic pain in mice [280], sodium channels may be the primary target that analgesics and analgesic adjuvants act on.

## 5. Actions of Plant-Derived Compounds on Nerve AP Conduction

Many of the plant-derived compounds activate the TRP channels located in the peripheral terminals of primary-afferent Aδ-fiber and C-fiber neurons, resulting in AP production, which in turn leads to temperature sensation and nociception. For example, capsaicin, allyl isothiocyanate, and menthol open the TRPV1, TRPA1, and TRP melastatin-8 (TRPM8) channels, respectively (for example, refer to [15,281,282]). On the other hand, TRPV1, TRPA1, and TRPM8 channels are located in primary-afferent neuron central terminals in the SG of the spinal dorsal horn; the central terminal TRP channels are activated by various plant-derived compounds such as capsaicin, allyl isothiocyanate, menthol, eugenol, carvacrol, thymol, (-)-carvone, (+)-carvone, 1,4-cineole, 1,8-cineole, (±)-linalool, and geraniol (refer to [283] and reviews [284,285]). This activation is thought to be involved in the modulation of excitatory and inhibitory synaptic transmissions in SG neurons, resulting in nociceptive transmission modulation. Consistent with this idea, excitatory and inhibitory transmissions in SG neurons are affected by a variety of endogenous pain modulators (for example, refer to [17]).

As with analgesics and analgesic adjuvants, CAPs in the frog sciatic nerve were inhibited by many of the plant-derived chemicals that cause analgesia when they topically, orally, intraperitoneally, or intrathecally were applied [286,287]. Carvacrol, thymol, citronellol, bornyl acetate, citral, citronellal, geranyl acetate, and geraniol reduced frog sciatic nerve CAP peak amplitudes with the $IC_{50}$ values of 0.34, 0.34, 0.35, 0.44, 0.46, 0.50, 0.51, and 0.53 mM, respectively (Table 1). Although capsaicin's $IC_{50}$ value was not able to be evaluated due to its lower solubility, capsaicin at 0.1 mM reduced CAP peak amplitudes by 36% (Table 1); this action could be attributed to at least part of the alleviation of chronic pain that is produced when capsaicin is applied to the skin ([288,289]; for example, refer to [15]). The activities of plant-derived compounds were similar to those of analgesic adjuvants and NSAIDs ([164,290–292]; refer to review [33]). Thus, their $IC_{50}$ values were comparable to values that duloxetine, amitriptyline, aceclofenac, tolfenamic acid, meclofenamic acid, and flufenamic acid have (0.23, 0.26, 0.47 0.29, 0.19, and 0.22 mM, respectively). Furthermore, (+)-pulegone, (-)-carvone, (+)-carvone, (+)-borneol, (±)-linalool, (-)-menthone, (+)-menthone, (-)-carveol, α-terpineol, rose oxide, cinnamaldehyde, and allyl isothiocyanate attenuated frog sciatic nerve CAP peak amplitudes with $IC_{50}$ values of 1.4, 1.4, 2.0, 1.5, 1.7, 1.5, 2.2, 1.3, 2.7, 2.6, 1.2, and 1.5 mM, respectively (Table 1), which were similar to those of fluoxetine and desipramine (1.5 and 1.6 mM, respectively). Linalyl acetate, eugenol, (-)-menthol, and (+)-menthol had $IC_{50}$ values of 0.71, 0.81, 1.1, and 0.93 mM, respectively

(Table 1); these values were close to those of diclofenac, maprotiline, and trazodone, which are 0.94, 0.95, and ca. 1.0 mM, respectively.

The cinnamaldehyde and allyl isothiocyanate activities were not affected by a non-selective TRP antagonist, ruthenium red. This result indicates that TRP channels are not involved in CAP inhibitions produced by them [291]. Capsaicin at high concentrations (0.03–0.1 mM) inhibited voltage-dependent sodium channels in rodents without TRPV1 channel activation [293–295]. Other plant-derived compound-induced CAP inhibitions are possibly caused by an attenuation of TTX-sensitive voltage-dependent sodium channels (for example, eugenol [296], thymol [297], carvacrol [298], and linalool [299]; refer to review [300]). The $IC_{50}$ value (0.37 mM) for carvacrol to inhibit voltage-dependent sodium channels in rat DRG neurons [301] was very similar to that (0.34 mM) for the frog sciatic nerve CAP inhibition.

Some plant-derived compounds had weak inhibitory effects on frog sciatic nerve CAPs. For example, 1,8-cineole, 1,4-cineole, zingerone, guaiacol, and vanillin had $IC_{50}$ values of 5.7, 7.2, 8.3, 7.7, and 9.0 mM, respectively. *p*-Cymene (2 mM), myrcene (5 mM), vanillylamine, and (+)-limonene (each 10 mM) reduced CAP peak amplitudes by 22, 7, 12, and 8%, respectively; vanillic acid (7 mM), *p*-menthane, and menthyl chloride (each 10 mM) did not affect CAPs (Table 1). Their compounds' activities were much smaller than those of NSAIDs and analgesic adjuvants. In conclusion, some plant-derived compounds can be used instead of NSAIDs and analgesic adjuvants with respect to nerve AP conduction inhibition. Plant-derived compounds are expected to have side effects less than synthetic analgesics.

Although the above-mentioned plant-derived compounds have open chains or six-membered rings, a seven-membered ring chemical, hinokitiol (β-thujaplicin; 2-hydroxy-4-isopropylcyclohepta-2,4,6-trien-1-one, which is present in essential oils obtained from cypress trees [302]), also concentration-dependently attenuated the peak amplitudes of CAPs recorded from the frog sciatic nerve, and an $IC_{50}$ value for this activity was 0.54 mM (Table 1). This value was similar to those of many other plant-derived compounds. The CAP suppression is probably mediated by certain interaction in which the carbonyl (-C=O), isopropyl (-CH(CH$_3$)$_2$), and hydroxyl (-OH) groups of hinokitiol are involved [303]. The -C=O group of hinokitiol helps its seven-membered ring act as a benzene ring, while the -CH(CH$_3$)$_2$ and -OH groups are pivotal for CAP suppression caused by hinokitiol. This idea is in favor of the observation that benzene ring compounds that have -CH(CH$_3$)$_2$ and -OH groups, including thymol, carvacrol, biosol (the last of which is a stereoisomer of thymol and carvacrol, and has an $IC_{50}$ value of 0.58 mM), and 4-isopropylphenol (0.85 mM), can suppress frog sciatic nerve CAPs (refer to above; [290,303]). As with other plant-derived compounds, hinokitiol has a variety of actions, including suppression of apoptosis [304], the activities of anti-bacterium, anti-inflammation [305], insecticide [306], anti-fungus [307], anti-tumor [308], and cytotoxicity [309,310]. Hinokitiol, which is used as a skin medicine for inflammation reduction, may have a local anesthetic action. Applying a hinokitiol-containing oral care gel to the oral mucosa is reported to relieve oral pain in patients having oral lichen planus related to the infection of hepatitis C virus [311]. Such a pain attenuation is likely to be partly mediated by hinokitiol's local anesthetic effect.

A CAP inhibitory action similar to hinokitiol was seen with a general anesthetic, propofol (2,6-diisopropylphenol; [312–314]; refer to reviews [315,316]), which has two -CH(CH$_3$)$_2$ groups and one -OH group attached to the benzene ring. Thus, propofol concentration-dependently reduced CAP peak amplitudes in the frog sciatic nerve and this activity had an $IC_{50}$ of 0.14 mM, which is smaller than that of hinokitiol (Table 1; [29]). Consistent with this experimental result, propofol reportedly suppressed APs that were extracellularly recorded in the human and mammalian CNS [317,318]. Such an inhibitory action of propofol on nerve AP conduction may contribute to its antinociceptive effect together with a propofol-induced enhancement of GABA$_A$ receptor responses in SG neurons [319].

In clinical practice, traditional Japanese medicines (called Kampo medicines) composed of crude chemicals isolated from plants are prescribed for various purposes such as

analgesia along with Western medicines in Japan [320–325]. Daikenchuto is one of most frequently prescribed Kampo medicines and is used to treat cold sensation and dysmotility in the abdomen. Frog sciatic nerve CAP amplitudes were concentration-dependently reduced by daikenchuto and by other Kampo medicines, such as rikkosan, kikyoto, rikkunshito, shakuyakukanzoto, and kakkonto. Among these medicines, daikenchuto had the greatest effect and an $IC_{50}$ value for this effect was 1.1 mg/mL. Daikenchuto contains three kinds of crude medicine, such as Japanese pepper, processed ginger, and ginseng radix. The former two have been shown to suppress CAPs while the last has no effects on CAPs. Japanese pepper had an $IC_{50}$ value of 0.77 mg/mL, and processed ginger at a concentration of 2 mg/mL reduced the peak amplitudes of CAPs by 31% [326]. A small part of the analgesic effect of Kampo medicine may be due to its nerve AP conduction inhibitory action.

The frog sciatic nerve CAP inhibitory action of plant-derived compounds is reversible or irreversible, depending on the type of compound. Therefore, in clinical practice, plant-derived compounds administered to peripheral nerves may or may not show tolerance in nerve AP conduction inhibition, depending on the type of compound.

**Table 1.** Effects of clinical analgesics, analgesic adjuvants, and plant-derived chemicals on fast-conducting frog sciatic nerve CAPs.

| Category | Compound | $IC_{50}$ (mM) | Studied Concentration (mM) | Observed AP Reduction (%) | References |
|---|---|---|---|---|---|
| Acetic acid-based NSAIDs | Diclofenac | 0.94 | - | - | [26] |
| | Aceclofenac | 0.47 | - | - | [26] |
| | Indomethacin | - | 1 | 38 | [26] |
| | Acemetacin | - | 1 | 38 | [26] |
| | Etodolac | - | 1 | 15 | [26] |
| | Sulindac | - | 1 | 0 | [26] |
| | Feblinac | - | 1 | 0 | [26] |
| Fenamic acid-based NSAIDs | Tolfenamic acid | 0.29 | - | - | [26] |
| | Meclofenamic acid | 0.19 | - | - | [26] |
| | Mefenamic acid | - | 0.2 | 16 | [26] |
| | Flufenamic acid | 0.22 | - | - | [26] |
| Salicylic acid-based NSAID | Aspirin | - | 1 | 0 | [26] |
| Propionic acid-based NSAIDs | Ketoprofen | - | 1 | 0 | [26] |
| | Naproxen | - | 1 | 0 | [26] |
| | Ibuprofen | - | 1 | 0 | [26] |
| | Loxoprofen | - | 1 | 0 | [26] |
| | Flurbiprofen | - | 1 | 0 | [26] |
| Enolic acid-based NSAIDs | Meloxicam | - | 0.5 | 0 | [26] |
| | Piroxicam | - | 1 | 0 | [26] |
| Opioids | Tramadol | 2.3 | - | - | [27] |
| | Mono-*O*-desmethyl-tramadol | - | 5 | 9 | [27] |
| | Morphine | - | 5 | 15 | [28] |
| | Codeine | - | 5 | 30 | [28] |
| | Ethylmorphine | 4.6 | - | - | [28] |
| Amide-type local anesthetics | Lidocaine | 0.74 | - | - | [28] |

**Table 1.** *Cont.*

| Category | Compound | IC$_{50}$ (mM) | Studied Concentration (mM) | Observed AP Reduction (%) | References |
|---|---|---|---|---|---|
| | Ropivacaine | 0.34 | - | - | [27] |
| | Prilocaine | 1.8 | - | - | [67] |
| | Levobupivacaine | 0.23 | - | - | [30] |
| | Bupivacaine | - | 0.5 | 76 | [30] |
| Ester-type local anesthetics | Cocaine | 0.80 | - | - | [28] |
| | Procaine | 2.2 | - | - | [164] |
| | Benzocaine | 0.80 | - | - | [29] |
| | Tetracaine | 0.014 | - | - | [32] |
| Other-type local anesthetic | Pramoxine | 0.21 | - | - | [67] |
| Antiepileptics | Lamotrigine | 0.44 | - | - | [30] |
| | Carbamazepine | 0.50 | - | - | [30] |
| | Oxcarbazepine | - | 0.5 | 20 | [30] |
| | Phenytoin | - | 0.1 | 15 | [30] |
| | Gabapentin | - | 10 | 0 | [30] |
| | Topiramate | - | 10 | 0 | [30] |
| | Sodium valproate | - | 10 | 0 | [30] |
| Antidepressants | Duloxetine | 0.23 | - | - | [31] |
| | Fluoxetine | 1.5 | - | - | [31] |
| | Amitriptyline | 0.26 | - | - | [31] |
| | Desipramine | 1.6 | - | - | [31] |
| | Maprotiline | 0.95 | - | - | [31] |
| | Trazodone | ca. 1.0 | - | - | [31] |
| Adrenoceptor agonists | Adrenaline | - | 1 | 0 | [32] |
| | Noradrenaline | - | 1 | 0 | [32] |
| | Dexmedetomidine | 0.40 | - | - | [32] |
| | Oxymetazoline | 1.5 | - | - | [32] |
| | Clonidine | - | 2 | ca. 20 | [32] |
| | Phenylephrine | - | 1 | 0 | [32] |
| | Isoproterenol | - | 1 | 0 | [32] |
| Open chain or six-membered plant-derived compounds | Carvacrol | 0.34 | - | - | [290] |
| | Thymol | 0.34 | - | - | [290] |
| | Citronellol | 0.35 | - | - | [292] |
| | Bornyl acetate | 0.44 | - | - | [292] |
| | Citral | 0.46 | - | - | [292] |
| | Citronellal | 0.50 | - | - | [292] |
| | Geranyl acetate | 0.51 | - | - | [292] |
| | Geraniol | 0.53 | - | - | [292] |
| | Capsaicin | - | 0.1 | 36 | [164] |
| | (+)-Pulegone | 1.4 | - | - | [290] |
| | (-)-Carvone | 1.4 | - | - | [290] |
| | (+)-Carvone | 2.0 | - | - | [290] |
| | (+)-Borneol | 1.5 | - | - | [292] |
| | (±)-Linalool | 1.7 | - | - | [292] |

**Table 1.** *Cont.*

| Category | Compound | $IC_{50}$ (mM) | Studied Concentration (mM) | Observed AP Reduction (%) | References |
|---|---|---|---|---|---|
| | (-)-Menthone | 1.5 | - | - | [290] |
| | (+)-Menthone | 2.2 | - | - | [290] |
| | (-)-Carveol | 1.3 | - | - | [290] |
| | $\alpha$-Terpineol | 2.7 | - | - | [292] |
| | Rose oxide | 2.6 | - | - | [292] |
| | Cinnamaldehyde | 1.2 | - | - | [291] |
| | Ally isothiocyanate | 1.5 | - | - | [291] |
| | Linalyl acetate | 0.71 | - | - | [292] |
| | Eugenol | 0.81 | - | - | [164] |
| | (-)-Menthol | 1.1 | - | - | [290] |
| | (+)-Menthol | 0.93 | - | - | [290] |
| | 1,8-Cineole | 5.7 | - | - | [290] |
| | 1,4-Cineole | 7.2 | - | - | [290] |
| | Zingerone | 8.3 | - | - | [164] |
| | Guaiacol | 7.7 | - | - | [164] |
| | Vanilin | 9.0 | - | - | [164] |
| | *p*-Cymene | - | 2 | 22 | [292] |
| | Myrcene | - | 5 | 7 | [292] |
| | Vanillylamine | - | 10 | 12 | [164] |
| | (+)-Limonene | - | 10 | 8 | [290] |
| | Vanillic acid | - | 7 | 0 | [164] |
| | *p*-Menthane | - | 10 | 0 | [290] |
| | Menthyl chloride | - | 10 | 0 | [290] |
| Seven-membered ring plant-derived compound | Hinokitiol | 0.54 | - | - | [303] |
| General anesthetic | Propofol | 0.14 | - | - | [29] |

The $IC_{50}$ value for CAP inhibition and extent of CAP amplitude reduction at each concentration are shown here.

## 6. Conclusions

This entry demonstrated that frog sciatic nerve CAPs are depressed by some NSAIDs, analgesic adjuvants, and plant-derived compounds that have analgesic activities with similar efficiencies, as well as opioids that have lower efficiencies than the above-mentioned categories. Although the CAPs are derived from TTX-sensitive APs generated in fast-conducting A$\alpha$-fibers, nociceptive message is transferred through slow-conducting A$\delta$-fibers and C-fibers [1]. Slow-conducting frog sciatic nerve CAPs could not be recorded, as A$\delta$-fiber CAPs were not able to be separated from A$\alpha$-fiber ones. The peak amplitude and conduction velocity of C-fiber CAP were much smaller than those of fast-conducting A$\alpha$ ones [25]. Therefore, it was not possible to investigate how slow-conducting CAPs are affected by the antinociceptive compounds. In order to know details regarding the differences in the magnitude of nerve AP conduction suppression among a variety of analgesic compounds, it would be required to investigate how slow-conducting CAPs are affected by the compounds.

In nerves other than the sciatic nerve in frogs, antinociceptive drugs have reportedly suppressed both A-fiber and C-fiber CAPs. For instance, in the vagus nerve in rabbits, fentanyl and sufentanil inhibited C-fiber CAPs with magnitudes less than those observed in A-fiber ones [85], and lidocaine suppressed both myelinated A-fiber and unmyelinated

C-fiber nerve AP conduction ([327]; lidocaine inhibited rat A-fiber CAPs more effectively than C-fiber CAPs [328]). Clonidine produced both A$\alpha$-fiber and C-fiber CAP amplitude reductions (refer to Section 3.4). Furthermore, many researchers have demonstrated a suppression by NSAIDs [56,63,329], lidocaine, and $\alpha_2$-adrenoceptor agonists [145] of TTX-resistant voltage-dependent sodium channels, which may be involved in slow-conducting CAP production. Knocking down Nav1.8 channels resistant to TTX reportedly resulted in suppressed pain associated with neuropathy or inflammation in rats [330].

Because the analgesic and analgesic adjuvant concentrations needed for CAP inhibition are higher than clinically relevant ones (as mentioned above), nerve AP conduction inhibition may only occur when the drugs are administered near nerve fibers or are accumulated in the nervous system. If such a suppression is produced in A$\alpha$-fibers that innervate skeletal muscle, this would produce unwanted secondary effects, such as muscle paralysis. Thus, the drug will have to be used at the lowest concentration while it is effective. Primary-afferent C-fibers and A$\delta$-fibers have diameters smaller than those of A$\alpha$-fibers. Therefore, if the drugs act on voltage-dependent sodium channels from the cytoplasm side, nerve AP conduction would be inhibited in C-fibers earlier than in A$\alpha$-fibers because the surface-to-volume ratio differs between the fibers. Because at least part of the analgesic action of most agents relies on the suppression of nerve AP conduction mediated by voltage-dependent TTX-sensitive and TTX-resistant sodium channels, the careful use of these compound aids in the study of nociception as well as clinical pain relief.

**Funding:** This research received no external funding.

**Conflicts of Interest:** The author declares no conflict of interest.

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
