# Peer review of "Inhibitory Actions of Clinical Analgesics, Analgesic Adjuvants, and Plant-Derived Analgesics on Nerve Action Potential Conduction"

_encyclopedia, doi:10.3390/encyclopedia2040132_

Round 1

Reviewer 1 Report

This is a well written review paper. I have only a few minor corrections presented in the manuscript

Author Response

Response:

  1. As instructed by you, when only review articles are cited, I have left the article numbers and removed “see for review”. However, this word has been preserved when review articles are cited together with original articles.
  2. According to your instruction, I have put commas before and after “drug” throughout the manuscript.
  3. I am very grateful to you for pointing out English mistakes in the manuscript. I have corrected all mistakes that were pointed out.

Author Response

My revision and opinion in response to the comments of Reviewer #2:

Thank you very much for reviewing my manuscript and providing valuable comments for this revision. I would like to reply to your comments as follows:

General Recommendation: Major Revisions

General Comments: The author is to be commended on a thorough exploration of analgesic action from the standpoint of commonly used agents on the nervous system. Although some language editing will be required, the adaptation of the previous paper seems to be successful and the resulting Encyclopedia entry will be of great use for neuroscientists exploring ideas for pain relief research. However, the paper is not useful for general scientific interest and must contain at least some commentary to place known results in context. Rewording the language for accessibility to a general audience is a high priority for Encyclopedia as it is not a neuroscience-focused journal. Specific comments are below.

Specific Comments:

Section 1.0 Introduction:

  1. Lines 51-62: This section is jarring and deals primarily with the methodology of examining AP in isolation. It would be better if this section had a transition at the beginning, such as “The study of AP in higher-order mammals is complicated by the need to dissect out individual nerves to isolate them from peripheral stimulation. For this reason, neuroscience relies on nerve extraction from lower-order animals with conserved AP mechanisms, such as reptiles or insects.” Alternately, the frog nerve section could be marked by a header, such as “1.1 Isolation Methods for Testing Analgesic Action on Nerve Fibers”

Response:

Thank you very much for your instruction. As instructed by you, the frog nerve section has been marked by a header, “1.1 Isolation Methods for Testing Analgesic Action on Nerve Fibers”. This section begins with the following sentence given by you, “The study of AP in higher-order mammals is complicated by the need to dissect out individual nerves to isolate them from peripheral stimulation. For this reason, neuroscience relies on nerve extraction from lower-order animals with conserved AP mechanisms, such as insects, reptiles, squids or frogs (for example, see [23])”.

  1. Line 69: The word “many” is ambiguous and “diverse” is recommended to impart to the reader the idea that there are both lots of compounds we have discovered and even more that we haven’t.

Response:

Thank you very much for your instruction. The word “many” has been altered to “diverse”.

Section 2:

  1. Line 79: It might be better to say “NSAIDS globally downregulate nociception by multiple mechanisms…” since the idea of “producing” antinociception could mislead readers into thinking that some proteins, enzymes, or other unique compounds are created from the drug itself rather than the mostly suppressive or manipulative effects that NSAIDs use.

Response:

I appreciate you for your throughtful comment. As instructed by you, “NSAIDs produce antinociception by various mechanisms” has been changed to “NSAISs globally downregulate nociception by multiple mechanisms”.

  1. Line 177: A short section on the durations and persistence of NSAID effect, coupled with any knowledge on tolerance and safety (with respect to the nervous system or long-term changes to AP conductivity) would be useful here. Please try to write it in a simpler fashion for general scientific interest so that researchers in non-neuroscience fields could understand it and use it as a springboard for future research.

Response:

The sentence, “Although NSAID is known to exhibit a tolerance effect (reduced response to a drug after its repeated use), this effect could not be explained by NSAID-induced peripheral nerve AP conduction inhibition revealed here, because decreased frog sciatic nerve CAP by diclofenac or aceclofenac treatment recovers to almost control level after treatment”, has been added in the last of “NSAID section”.

  1. Line 194: Again, a short section on addiction might be useful here, as addictive properties may change nociceptive thresholds over time.

Response:

The sentences, “Since decreased frog sciatic nerve CAP by treatment with tramadol does not recover to control level after treatment, repeated tramadol application may not have any effect, resulting in a tolerance. On the other hand, it is unlikely that the nerve AP conduction inhibition revealed here is related to tramadol’s addiction, because addiction is generally thought to be due to an activation of reward centers in the brain”, have been added in the “Tramadol section”.

  1. Line 257: Again, a short section on clinical use might be useful.

Response:

The sentence, “Although opioids such as morphine, codeine and ethylmorphine are known to have a tolerance effect, this could not be explained by their inhibitory effects on nerve AP conduction, because their opioids inhibit frog sciatic nerve CAPs in a reversible manner [28]”, has been added in the last of ”Morphine, codeine and ethylmorphine section”.

  1. Lines 317-331: Excellent section on clinical usage. The other drugs deserve similar treatment…but those sections don’t need to be so long. Including addiction and any other relevant long-term effects (such as habituation or tolerance) is necessary since the paper did introduce the concept of “chronic” pain, which usually requires long-term medication or other methods of pain relief that change nociception over time.

Response:

The sentence, “Although opioids exhibit side effects such as tolerance and addiction, these effects appear to be due to the action on synaptic transmission in the CNS rather than nerve AP conduction, because synaptic transmission but not nerve AP conduction can be changed in efficacy plastically”, has been added in the last of “Opioid section”.

Sections 3, 4, and 5:

  1. Alternately, a paragraph or two at the end of these sections (and Section 2) could explain the required clinical aspects of use, addiction, and nociceptive changes over time in lieu of a few short sentences at the end of each subsection. Such a paragraph is needed here to place results in a context that makes the information useful to clinicians, even if they aren’t the primary audience.

Response:

The end of local anesthetic section:

The sentence, “In clinical practice, local anesthetics administered locally to peripheral nerves may be repeatedly effective without being tolerated, because both amide-type and ester-type local anesthetics reversibly inhibit frog sciatic nerve CAPs”, has been added.

The end of antiepileptic section:

The sentence, “In clinical practice, antiepileptics administered locally to peripheral nerves may be repeatedly effective without being tolerated, because the inhibitory actions of antiepipeptics on frog sciatic nerve CAPs are partially but reversible”, has been added.

The end of antidepressant section:

The sentence, “In clinical practice, antidepressants administered locally to peripheral nerves may be repeatedly effective without being tolerated, because the inhibitory actions of antidepressants on frog sciatic nerve CAPs are partially but reversible”, has been added.

The end of adrenoceptor section:

The sentence, “DEX administered locally to peripheral nerves may be repeatedly effective without being tolerated, because DEX reversibly inhibits frog sciatic nerve CAPs”, has been added.

The end of plant-derived compound section:

The sentences, “The frog sciatic nerve CAP inhibitory action of plant-derived compounds is reversible or irreversible, depending on the type of compound. Therefore, in clinical practice, plant-derived compounds administered to peripheral nerves may or may not show tolerance in nerve AP conduction inhibition, depending on the type of compound”, have been added.

  1. Table 1 is welcome but should be redone to avoid confusion. An example is given below.

Please make the entire table in this way to clarify results

Response:

According to your instruction, I have revised Table 1, where there is an IC50 column in addition to the columns indicated by you.

Section 6: Conclusion

  1. Lines 910-921: This section is good but redundant from a clinical perspective. Mentioning the need for titration and then transitioning into Lines 919-922 would be ideal. Suggested: “Since at least part of the analgesic action of most agents relies on suppression of nerve AP conduction mediated by voltage-gated, TTX-sensitive and resistant Na+ channels, careful use of these compounds both aid in the study of nociception as well as clinical pain relief.”

Response:

Thank you very much for the suggested sentence. “At least part of analgesia produced by analgesics and analgesic adjuvants may be due to their suppressive effects on nerve AP conduction that is mediated by voltage-gated TTX-sensitive and TTX-resistant voltage-gated Na+ channel activations” in my manuscript has been changed to “Since at least part of the analgesic action of most agents replies on suppression of nerve AP conduction mediated by voltage-gated TTX-sensitive and TTX-resistant Na+ channels, careful use of these compounds aids in the study of nociception as well as clinical pain relief”.

Reviewer 3 Report

The entry titled “Inhibitory Actions of Analgesics, Analgesic Adjuvants and Plant-Derived Analgesics on Nerve Action Potential Conduction” summarize data on the inhibitory effects of clinically-used analgesics, analgesic adjuvants and plant-derived analgesics on fast-conducting CAPs. The topic is interesting, manuscript is well written and has a good readability. Just minor remarks for the text:

Line 70. “This review article will describe the effects of their chemical compounds” – it is not clear what is about. It is better to rephrase.

Line 84. “and an involvement of opioids and endocannabinoids” – in such phrase it is about drug types but in the manuscript mechanisms of action are described, so it should be corrected.

Line 114. “[]” instead of “()”

Line 165. “in the rat exocrine pancreas” – may be “exocrine” is not needed here?

Line 198. “is a clinically used opioid administered orally in the CNS” – strange phrase, it better to rewrite it.

Line 267. “coceine” instead of codeine

Line 282. “Consistent with this idea, a potency of rat sciatic nerve CAP inhibition was in the order of isopropylcocaine > cocaethylene > cocaine [120]” – it is not clear the purpose of this sentence in this place since the topic of the section is opioid analgesics.

Line 320. “Opioids administered centrally can act on the PNS as well as the CNS, because opioids are transported from the brain to the periphery by P-glycoprotein [130]. Supporting this idea, subcutaneous administration of blood brain barrier-impermeable N-methyl-morphine produced antinociception in an acetic acid-writhing model in mice” – used explanation of the idea is not correct and does not involve P-glycoprotein in the mechanism of peripherally action.

Line 329. “Peripherally-applied codeine might have a similar effect to that of morphine, because codeine is metabolized to morphine via O-demethylation in humans and animals” – please explain the meaning of this since it is known that codeine has a similar to morphine pharmacological actions.

Line 761. Actually, codeine and morphine are also plant-derived compounds and they are placed to another section. Maybe it possible to change the chapter’s title?Line 70. “This review article will describe the effects of their chemical compounds” – it is not clear what is about. It is better to rephrase.

Line 84. “and an involvement of opioids and endocannabinoids” – in such phrase it is about drug types but in the manuscript mechanisms of action are described, so it should be corrected.

Line 114. “[]” instead of “()”

Line 165. “in the rat exocrine pancreas” – may be “exocrine” is not needed here?

Line 198. “is a clinically used opioid administered orally in the CNS” – strange phrase, it better to rewrite it.

Line 267. “coceine” instead of codeine

Line 282. “Consistent with this idea, a potency of rat sciatic nerve CAP inhibition was in the order of isopropylcocaine > cocaethylene > cocaine [120]” – it is not clear the purpose of this sentence in this place since the topic of the section is opioid analgesics.

Line 320. “Opioids administered centrally can act on the PNS as well as the CNS, because opioids are transported from the brain to the periphery by P-glycoprotein [130]. Supporting this idea, subcutaneous administration of blood brain barrier-impermeable N-methyl-morphine produced antinociception in an acetic acid-writhing model in mice” – used explanation of the idea is not correct and does not involve P-glycoprotein in the mechanism of peripherally action.

Line 329. “Peripherally-applied codeine might have a similar effect to that of morphine, because codeine is metabolized to morphine via O-demethylation in humans and animals” – please explain the meaning of this since it is known that codeine has a similar to morphine pharmacological actions.

Line 761. Actually, codeine and morphine are also plant-derived compounds and they are placed to another section. Maybe it possible to change the chapter’s title?

Author Response

My revision and opinion in response to the comments of Reviewer #3:

Thank you very much for reviewing my manuscript and providing valuable comments for this revision. I would like to reply to your comments as follows:

The entry titled “Inhibitory Actions of Analgesics, Analgesic Adjuvants and Plant-Derived Analgesics on Nerve Action Potential Conduction” summarize data on the inhibitory effects of clinically-used analgesics, analgesic adjuvants and plant-derived analgesics on fast-conducting CAPs. The topic is interesting, manuscript is well written and has a good readability. Just minor remarks for the text:

Line 70. “This review article will describe the effects of their chemical compounds” – it is not clear what is about. It is better to rephrase.

Response:

“chemical compounds” has been replaced by “antinociceptive drugs”.

Line 84. “and an involvement of opioids and endocannabinoids” – in such phrase it is about drug types but in the manuscript mechanisms of action are described, so it should be corrected.

Response:

“and an involvement of opioids and endocannabinoids” has been replaced by “opioid-receptor activation [54,55], cannabinoid-receptor activation and endocannabinoid level increase [56]”.

Line 114. “[]” instead of “()”

Response:

I am sorry but I cannot understand this correction. I think that “enolic acid-based NSAIDs [meloxicam (0.5 mM) and piroxicam (1 mM)] [26]” is OK with respect to the use of parentheses.

Line 165. “in the rat exocrine pancreas” – may be “exocrine” is not needed here?

Response:

I appreciate you for pointing this out. I have deleted “exocrine”.

Line 198. “is a clinically used opioid administered orally in the CNS” – strange phrase, it better to rewrite it.

Response:

Thank you very much for your pointing this out. This phrase has been changed to “is an orally-administrated opioid in the clinical setting that acts on the CNS”.

Line 267. “coceine” instead of codeine

Response:

This mistake has been corrected. Thank you very much.

Line 282. “Consistent with this idea, a potency of rat sciatic nerve CAP inhibition was in the order of isopropylcocaine > cocaethylene > cocaine [120]” – it is not clear the purpose of this sentence in this place since the topic of the section is opioid analgesics.

Response:

I appreciate you for pointing this out. I have changed “Consistent with this idea” into the following sentence, “Structure-activity relationships similar to that of opioids have been demonstrated for other chemicals. Thus,”.

Line 320. “Opioids administered centrally can act on the PNS as well as the CNS, because opioids are transported from the brain to the periphery by P-glycoprotein [130]. Supporting this idea, subcutaneous administration of blood brain barrier-impermeable N-methyl-morphine produced antinociception in an acetic acid-writhing model in mice” – used explanation of the idea is not correct and does not involve P-glycoprotein in the mechanism of peripherally action.

Response:

Thank you very much for your attention. I have deleted “Supporting this idea”.

Line 329. “Peripherally-applied codeine might have a similar effect to that of morphine, because codeine is metabolized to morphine via O-demethylation in humans and animals” – please explain the meaning of this since it is known that codeine has a similar to morphine pharmacological actions.

Response:

Thank you very much for your attention. I have changed “a similar effect to that of morphine” into “a nerve AP conduction inhibitory effect similar to morphine” to make the meaning clear.

Line 761. Actually, codeine and morphine are also plant-derived compounds and they are placed to another section. Maybe it possible to change the chapter’s title?

Response:

According to your suggestion, I have changed the section title from “Morphine, Codeine and Ethylmorphine” to “Other Opioid-Related Compounds (Morphine, Codeine and Ethylmorphine)”.

Round 2

Reviewer 1 Report

The author responted to all my comments and I do believe that the mamnuscript is fine. I have just a couple of suggestions, clearly indicated in the attached manuscript PDF file.

Author Response

My re-revision and opinion in response to the comments of Reviewer #1:

Thank you very much for reviewing my revised manuscript and providing valuable comments for this revision. I would like to reply to your comments as follows:

The author responted to all my comments and I do believe that the mamnuscript is fine. I have just a couple of suggestions, clearly indicated in the attached manuscript PDF file.

Response: 1.     Although "higher-order animals" is still used for mammals or vertebrates, and "lower-order animals" for (most of) the rest, these are rather anthropocentric and neither correct nor politically correct terms. Perhaps "poikilothermic" (including all the animals except mammals and birds) or "relatively simple animals" would be more appropriate than "lower-order"; and you could simply delete "higher-order'' and write just "mammals" Response:

Thank you very much for your idea. According to your advice, I have deleted "higher-order'' and written just "mammals". Moreover, “lower-order animals” has been changed to “relatively simple animals”.

  1. Although this depends on the particular Journal, I would prefer the British English "anaesthetics" and not the American English "anesthetics". It is also more correct when one considers the original Greek word "αναισθησια" for "anaesthesia".

Response:

Thank you very much for your idea. If this manuscript is revised according to the British English, the other words such as “neuron” and “fiber” also will have to be corrected, i.e., to “neurone” and “fibre”. Therefore, "anesthetics" is used as it is without changing to "anaesthetics".

  1. "NSAIDs are known".

Response:

As instructed by you, “NSAID is” has been corrected to “NSAIDs are”.

Reviewer 2 Report

Please forward on to language check.  All comments have been addressed.

Author Response

Please forward on to language check.  All comments have been addressed.

Thank you very much for reviewing my revised manuscript.

Reviewer 3 Report

The author improved the article according my remarks so it can be published.

Author Response

The author improved the article according my remarks so it can be published.

Thank you very much for reviewing my revised manuscript.

Round 3

Reviewer 1 Report

I have no further comments for the author. I do the believe the manuscript is fine.